# Effect of Work Stress and Eating Behavior: A Study Among Academicians in Türkiye

**DOI:** 10.3390/healthcare13141758

**Published:** 2025-07-20

**Authors:** Merve İnce-Palamutoğlu, Betül Oruçoğlu, Meltem İnce-Yenilmez, Gizem Ağır

**Affiliations:** 1Department of Nutrition and Dietetics, Faculty of Health Sciences, Afyonkarahisar Health Sciences University, Afyonkarahisar 03030, Türkiye; merve.palamutoglu@afsu.edu.tr (M.İ.-P.); betul.orucoglu@afsu.edu.tr (B.O.); 2Department of Economics, Faculty of Economics and Administrative Sciences, İzmir Democracy University, İzmir 35140, Türkiye; melteminceyenilmez@idu.edu.tr; 3Department of Nutrition and Dietetics, Faculty of Health Sciences, Acıbadem Mehmet Ali Aydınlar University, Istanbul 34752, Türkiye

**Keywords:** academicians, occupational stress, eating behavior, GWSS, AEQB

## Abstract

**Background/Objectives:** Occupational stress is a common issue among academics. This study aims to determine the level of work stress experienced by academics depending on their titles and the relationship between this stress and eating behaviors. **Methods:** The data for the study were collected through an online survey from 649 academicians working in universities in Türkiye between January and February 2025. This is a cross-sectional study, and the snowball sampling method was used to facilitate high participation rates. Data were collected using a demographic questionnaire, the General Work Stress Scale (GWSS) to assess work stress levels, and the Turkish version of the Adult Eating Behavior Questionnaire (AEBQ-TR) to evaluate eating behaviors. **Results:** Among the participants, 47.6% were of normal weight, with an average BMI of 25.85 ± 4.56 kg/m^2^. The median work stress score of male academicians (17.00) was significantly lower than that of female academicians (21.00) (*p* < 0.001). Similarly, the median eating behavior score was lower in male academicians (18.55) compared to females (19.78) (*p* < 0.001). Work stress levels decreased with increasing academic title, with professors reporting the lowest levels of stress (*p* < 0.001). **Conclusions:** The findings indicate that female academics are more likely to engage in emotional eating under stress, whereas male academics tend to alter their dietary preferences by avoiding certain foods. These results underscore the importance of stress management and the promotion of healthy eating habits among academics.

## 1. Introduction

Traditionally, universities fulfill the functions of education, basic scientific research, and service to society [1]. Academicians working at universities are responsible for producing knowledge and providing benefits by transferring this knowledge to society. In addition to teaching activities, academics are also responsible for carrying out research, scientific studies, and administrative duties. Academics are expected to have a constantly active and dynamic work tempo; however, these multifaceted responsibilities can create physical, psychosocial, and emotional burdens, causing academic stress [2].

The concept of stress, derived from the Latin words “*estrica*” and old French “*estrece*”, is as old as human history [3,4]. Stress is a psychological response that allows us to cope with problems and is a response to one or more events that produce negative emotions [3]. Stress is a common experience that individuals face throughout their lives. Studies reveal that approximately 33% of adults feel intense stress most days. While low-level stress is considered a physiological and psychological response that facilitates the body’s adaptation to environmental demands, this does not always lead to negative consequences. However, when stress persists and becomes chronic, it can have serious negative effects on individual health. Although everyone experiences stress from time to time, the types of stress differ in their effects [5]. According to Lazarus and Folkman’s (1984) Stress, Appraisal, and Coping Theory, there are two basic types of stress. These are called “Acute (Short Term) Stress and “Chronic (Long Term) Stress” [6]. Acute stress is a short-term form of stress, typically resulting from sudden stressors or challenging situations. In this case, the body exhibits temporary physiological changes, such as an increased heart rate and the release of adrenaline, along with the “fight or flight” response. In contrast, chronic stress occurs when stress persists for an extended period. Exposure to chronic stress can cause cumulative physiological and psychological effects over time, which can increase the risk of health problems such as cardiovascular disease, anxiety, and depression [7]. Klarrich (1993) stated that stress can cause problems such as fatigue, hypertension, and tension in individuals and occurs most frequently in work environments [8]. Occupational stress is defined as the harmful physical and emotional reactions that occur when the work’s requirements do not align with the employee’s abilities, resources, or needs [9]. According to the WHO (2022), each year the global economy loses 12 billion workdays at a cost of about $1 trillion due to stress, depression, and anxiety [10].

The effect of stress on eating behaviors may vary depending on the individual’s emotional state, past eating habits, and the type of stress they are exposed to [11]. In times of crisis (e.g., in a war environment), it has been found that perceived high stress is generally associated with individuals eating less [12]. In contrast, individuals experiencing moderate or mild stress have been observed to increase their consumption of high-calorie foods [11]. It has been found that emotional eating behavior increases as a coping mechanism after crisis situations such as natural disasters, especially in women [13]. The prolonged exposure to stress can profoundly affect one’s mental and physical well-being and affect various aspects of individuals’ lives, such as eating behaviors and energy level [14]. Stress contributes directly or indirectly to an increased appetite and energy intake and decreased physical activity [15]. Therefore, this effect on the mechanisms of energy intake and expenditure is associated with the etiology of obesity. It is known that the relationship between stress and food intake is highly individual [16]. In cases where stress is experienced chronically, the individual may increase food intake in response to stress and experience weight gain through a positive energy balance. On the contrary, they may decrease food intake and experience weight loss through a negative energy balance. A meta-analysis on this topic suggested that stress affects not only the quantity but also the quality of food intake. Under stress, individuals tend to consume foods that are perceived as more palatable, energy-dense, and more rewarding. This shift in preference is thought to be linked to the physiological role of such foods in moderating the acute effects of stress [17]. The reason for this is the activation of the hypothalamic–pituitary–adrenal (HPA) axis under stress and the release of stress hormones such as cortisol [18]. These hormones increase the desire for high-energy, delicious foods by affecting the brain’s reward processing and motivation pathways [16]. On the other hand, it has been determined that not only are female-gender-role stress associated with eating disorders, but also general stress levels; eating disorders are primarily linked to perceived life stress [19,20].

The high expectations and intense workload of academicians’ environments affect individuals’ mental and physical health and shape their quality of life. This study aimed to determine the relationship between the occupational stress experienced by academics due to their titles and their eating behaviors. This study is important because it is one of the limited number of studies examining the relationship between occupational stress and eating behaviors on academics in Türkiye. The findings to be obtained may shed light on individual and institutional interventions aimed at developing stress coping strategies and healthy life habits in the academic community.

## 2. Materials and Methods

### 2.1. Study Design, Sample Size, and Sampling

This is a descriptive, cross-sectional, exploratory study. Data for this study were collected through an online (Google Forms) questionnaire administered to 649 academicians working in higher education institutions of the Republic of Türkiye between January and February 2025. To ensure high participation, we used a snowball sampling method. Participants represented diverse academic ranks (e.g., research assistants, lecturers, associate professors, and professors), disciplines (e.g., social sciences, natural sciences, health sciences), and institutional types (public vs. private). The Electronic Document Management System (EDMS), used in both public and private institutions, is an integrated software system that enables the creation, signing, transmission, and archiving of internal and external correspondence in digital format. In this study, the EDMS was utilized to distribute announcements and reach participants within the scope of the snowball sampling method. In accordance with the Personal Data Protection Law (Law No. 6698) in Türkiye, no personally identifiable information (e.g., names, contact details, and institutional affiliations) was requested from participants. Prior to participation, individuals were informed about the purpose, scope, and procedures of the study, and electronic informed consent was obtained via the Google Forms platform.

According to data from the Republic of Türkiye Higher Education Institution, as of December 2024, there were 39,454 professors, 25,697 associate professors, 45,457 assistant professors, 35,207 lecturers, and 38,967 research assistants, totaling 184,000,782 academicians. According to the sample calculation, the sample had to include at least 383 participants with a 95% confidence interval and 5% acceptable error [21]. This study collected data from 649 academicians who voluntarily agreed to participate. A post hoc power analysis was conducted using G*Power 3.1 to determine the adequacy of the sample size. Assuming a medium effect size (w = 0.30), and a significance level of α = 0.05, the achieved statistical power was calculated to be approximately 0.996. This result indicates that the sample size was sufficient for making reliable and valid statistical inferences, even within a large population. We excluded participants who did not consent to participating in the study, who had a diagnosed eating disorder, and who were taking psychiatric medications.

### 2.2. Assessment Tools

#### 2.2.1. Ad Hoc Socio-Demographic Questionnaire

The data of the study were obtained through a questionnaire developed by the researchers following a literature review. The first section of the questionnaire consists of 17 items related to the socio-demographic characteristics of the participants. These questions cover variables such as gender, age, height, weight, academic title, income level, monthly food expenditure, diagnosed health conditions, and adherence to any dietary programs. Additionally, the questionnaire includes items regarding adequate and balanced nutrition status, the frequency of main and snack meal consumption, meal skipping behavior, and the reasons for skipping meals. Body Mass Index (BMI) levels were calculated using information about individuals’ body weight (kg) and height squared (m^2^). The rest consisted of the Adult Eating Behavior Questionnaire and the General Work Stress Scale.

#### 2.2.2. General Work Stress Scale (GWSS)

The validity and reliability of the Turkish version of the General Work Stress Scale, developed by De Bruin (2006) [22], was performed by Teleş (2021). The general work stress scale consists of 9 items and is a 5-point Likert type. A low score (1 or close to 1) on the scale items or the Scale General Average means that the level of work stress is low, and a high score (5 or close to 5) indicates that the level of work stress is high [3].

#### 2.2.3. The Adult Eating Behavior Questionnaire (AEBQ-TR)

AEBQ was developed by Hunot et al. (2016) [23] to evaluate the eating behavior of individuals. The reliability and validity of the AEBQ in Turkish was performed by Yardımcı et al. (2022) [24]. AEBQ is a scale developed to assess adult appetite characteristics. The adult eating behavior scale items consisting of thirty-five questions and seven subscales (Enjoyment of food (items 1, 3, and 4), Emotional over-eating (items 5, 8, 10, 16, and 21), Emotional under-eating (items 15, 18, 20, 27, and 35), Food fussiness (items 2, 7, 12, 19, and 24), Slowness in eating (items 14, 25, 26, and 29), Hunger + food responsiveness (items 6, 9, 13, 17, 22, 28, 32, 33, and 34), and Satiety responsiveness (items 11, 23, 30, and 31)) are classified with four food approaches and three food avoidance sub-dimensions. The food approach subscales capture behavioral tendencies that may contribute to increased energy intake. Among these, the hunger subscale reflects an individual’s sensitivity to physiological cues of hunger. Food responsiveness measures the tendency to react to external food cues such as the sight or smell of food. Emotional over-eating represents a behavioral inclination to consume more food in response to negative emotional states. Lastly, enjoyment of food reflects the degree of pleasure and positive attitude experienced toward eating. Collectively, these traits are considered to promote a higher drive to eat in response to internal or external stimuli. On the other hand, the food avoidance subscales are generally associated with behaviors that may contribute to reduced energy intake. Satiety responsiveness refers to an individual’s ability to recognize fullness cues and to stop eating accordingly. Emotional under-eating describes a pattern of decreased food intake during emotional distress, such as anxiety or sadness. Food fussiness denotes selective eating behaviors and reluctance to try unfamiliar foods. Finally, slowness in eating is characterized by a slower eating pace, which has been associated with increased awareness of satiety and reduced total intake. Together, these avoidance-related traits may serve as protective factors against overconsumption. Items 12, 14, 19, and 24 in the scale are among the reverse-coded questions. The scale is a 5-point Likert type. Each subscale consists of 3–5 questions. For each subscale total score, the scores of the questions forming the dimension are added, and the average is calculated. The total survey score is calculated by adding all subscale scores [24]. A high total AEBQ score indicates that the participant generally has high behavioral appetite characteristics, also known as “appetitive traits”. High scores on the subscales of the Food Approach dimension—Hunger, Food Responsiveness, Emotional Over-eating, and Enjoyment of Food—indicate a strong appetite and a greater tendency to eat in response to emotional states or external cues. Conversely, low scores on the Food Avoidance subscales—Satiety Responsiveness, Emotional Under-eating, Food Fussiness, and Slowness in Eating—may reflect lower sensitivity to satiety signals, reduced food intake during negative emotional states, and faster eating behavior [23].

### 2.3. Statistical Analysis

Data analysis examining the effects of work-related stress on the eating behaviors of academicians was performed using IBM SPSS Statistics version 26 (IBM Corp., Armonk, NY, USA). As a result of the normality analysis tests (Kolmogorow–Smirnov and Shapiro–Wilk), it was determined that GWSS, AEBQ-TR, and its sub-dimensions (Enjoyment of food, Emotional over-eating, Emotional under-eating, Food fussiness, Slowness in eating, Hunger + food responsiveness, and Satiety responsiveness) did not show a normal distribution because the “Sig.” values were less than 0.05. According to these results, non-parametric tests were preferred in the study. The median value was used for the descriptive statistics of the scales and their sub-dimensions. The level and direction of correlation between the scales were investigated using Spearman correlation analysis. In addition, the Mann–Whitney U test and the Kruskal–Wallis test were preferred for examining the differences in the scales according to demographic variables. All these statistical analyses were performed at a 95% confidence interval and with a *p*-value of 0.05 as a reference.

### 2.4. Ethical Considerations

Ethics approval was obtained from the Afyonkarahisar Health Sciences University Non-Interventional Scientific Research Ethics Committee (dated 3 January 2025, no. 2025/12). Additionally, consent for this study was obtained from the involved establishments. Written informed consent was obtained from the participants before this study started.

## 3. Results

A total of 649 academicians participated in this study, and 61.2% (*n* = 397) were female. The mean age of the participants was 40.96 ± 9.06 (min 24–max 66) years. Among the participants, the mean body mass index was 25.9 ± 4.6 kg/m^2^ (min 16.7–max 44.1). Table 1 analysis reveals that most academicians participating are Assistant Professors (26.5%) and Associate Professors (18.2%). Regarding monthly food expenditures as a percentage of revenue, 290 participants (44.7%) reported spending 25–50%, while 261 (40.2%) indicated spending, for food, 0–25%. Furthermore, 421 academicians (64.8%) reported having no health problems. Many participants (480; 74.0%) said they had never followed a diet program. When asked about their dietary habits, 313 academicians (48.2%) considered their nutrition adequate and balanced, whereas 171 (26.3%) believed it was inadequate, and 165 (25.5%) were unsure.

Of the academics participating in the study, 415 (63.9%) consume two main meals. 245 (37.7%) of the participants reported skipping a meal. The majority of participants skipped main meals because they lacked time (47.6%). Of the academicians, 210 (32.4%) reported that they did not snack regularly, while 362 (82.5%) stated that they occasionally skipped snacks. The most important reason for skipping snacks was a lack of time (*n* = 176, 40.1%) (Table 2).

### 3.1. GWSS and AEBQ-TR Total Scores

In the study, the overall median GWSS score was determined as 2.22 ± 0.90 (min: 9.00–max: 45.00). The total median AEBQ-TR score was determined as 19.26 ± 2.29 (min: 11.48–max: 30.60). The median scores of the AEBQ-TR subscales were defined as follows: Enjoyment of Food 4.00 ± 0.76, Emotional Over-Eating 2.60 ± 1.25, Emotional Under-Eating 2.80 ± 1.13, Food Fussiness 2.20 ± 0.80, Slowness in Eating 2.50 ± 1.14, Hunger + Food Responsiveness 2.56 ± 0.71, and Satiety Responsiveness 2.00 ± 0.60.

### 3.2. Gender Differences in Eating Behaviors and Work Stress

The Mann–Whitney U test revealed the differences between male and female academicians’ general work stress, adult eating behavior, and subscale scores. When Table 3 is examined, a statistically significant difference was found between the two groups regarding academicians’ work stress scores (U = 39,856.5, *p* < 0.001). The table shows that the work stress average of male academicians (Median = 17.00) is significantly lower than that of female academicians (Median = 21.00). A statistically significant difference was found between the two groups in terms of academicians’ eating behaviors (U = 35,463.5, *p* < 0.001). The table shows that the average eating behavior scores of male academicians (Median = 18.55) are significantly lower than those of female academicians (Median = 19.78). A higher AEBQ score indicates a higher tendency to eat or to eat more under the influence of emotional or external stimuli. Thus, male academicians’ lower AEBQ scores may suggest that their eating behaviors may be more suppressed or less sensitive to internal appetite cues. In contrast, female academics’ higher scores may indicate that they are more sensitive to food stimuli and more reactive in terms of food cravings.

There was no significant difference in the enjoyment of food and emotional under-eating subscale scores between genders (U = 49, 538.5, *p* > 0.05; U = 49, 429.0, *p* > 0.05, respectively). A significant difference was found between the scores they obtained from the emotional over-eating, food fussiness, slowness in eating, hunger + food responsiveness, and satiety responsiveness subscales and their gender (U = 38,840.0, *p* < 0.05; U = 44,486.0, *p* < 0.05; U = 41,619.5, *p* < 0.05; U = 42,349.0, *p* < 0.05; U = 37,382.5, *p* < 0.05, respectively). The mean enjoyment of food scores of the academicians were found to be the same in terms of gender. While the mean scores of male academicians on emotional under-eating and food fussiness subscales were higher than those of female academicians, the mean scores of female academicians on other subscales were higher (emotional under-eating, slowness in eating, hunger + food responsiveness, and satiety responsiveness). This situation can be interpreted as male academicians being able to suppress or control their eating behaviors when faced with negative emotional states.

### 3.3. Academicians Differences in Eating Behaviors and Work Stress

The Kruskal–Wallis’s test revealed the differences in general work stress and eating behaviors among academicians with different titles. When Table 4 is examined, a statistically significant difference was found between seven academic titles in terms of the work stress scores of academicians (H = 50.562, *p* < 0.001). As a result of the Tukey HSD multiple-comparison test, which was conducted to determine which title groups this difference was between (in other words, which groups the difference originated from), work stress scores significantly decrease as the academic title increases. Work stress scores are lower in the professor doctor than in other academicians. In addition, work stress scores are lower in associate professor academicians than in the research assistants’ doctor and research assistants.

A statistically significant difference was found among seven academic titles in terms of the adult eating behavior scores of academicians (H = 25.656, *p* < 0.001). As a result of the Tukey HSD multiple-comparison tests conducted to determine which title groups this difference is between (in other words, which groups the difference originates from), it was found that the difference originates between professor and doctor academicians (Median = 18.37) and research assistant doctors (Median = 20.20), lecturers (Median = 19.71), and research assistants (Median = 19.91). Table 4 also shows that no significant difference was found between the academicians’ scores on the subscales of food enjoyment, excessive emotional eating, emotional under-eating, picky eating, slow eating, hunger food sensitivity, and satiety sensitivity and their academic titles (*p* > 0.05) (Figure 1).

### 3.4. Work Stress–Eating Behaviors Relationship

The relationship between work stress and eating behaviors was evaluated using a Spearman correlation analysis. The correlation coefficient was interpreted as follows: 0.00 (no relationship), 0.01–0.29 (low-level relationship), 0.30–0.70 (moderate relationship), 0.71–0.99 (high-level relationship), and 1.00 (perfect relationship) [25]. When Table 5 is examined, low-level positive correlations were found between the GWSS and AEBQ-TR score (r = 0.214, *p* < 0.05). In other words, statistically weak but significant relationships were identified between the overall work stress scores and the total score of the AEBQ-TR, as well as the subscales of Emotional Over-eating and Hunger + Food Responsiveness.

A low positive correlation was found between academicians’ work stress and excessive emotional eating scores (r = 0.272, *p* < 0.05). A moderate positive correlation was found between academicians’ work stress and hunger + food sensitivity score (r = 0.301, *p* < 0.05). The analysis revealed no significant correlation between academicians’ work stress and their enjoyment of eating, emotional undereating, picky eating, or satiety sensitivity scores (r = 0.045, *p* > 0.05; r = −0.054, *p* > 0.05; r = 0.061, *p* > 0.05; r = −0.064, *p* > 0.05; and r = −0.032, *p* > 0.05, respectively) (Table 5).

## 4. Discussion

Acute stress initially suppresses appetite. Acute stress is mediated by the corticotropin-releasing hormone (CRH), which prepares the body for the “fight or flight” response. However, once the stress is over, glucocorticoid hormones (primarily cortisol) increase appetite. This can lead to excessive, high-calorie food consumption, especially when occupational stress becomes chronic. Stress increases the desire for “rewarding” foods, especially those high in fat and sugar. Such foods can both reduce feelings of stress by suppressing the activation of the hypothalamic–pituitary–adrenal (HPA) axis and can also cause weight gain [26]. Occupational stress is a significant issue for both employees and organizations, as it affects mental health, well-being, and productivity. Studies support the claim that mindfulness training benefits work engagement, reduces burnout, enhances leadership, improves productivity, and enhances cognitive function. Occupational stress impacts work performance, including decision-making, problem-solving, and productivity [27]. Occupational stress is emerging as a serious problem for research managers and academic staff, as it is for many other professions today. In this context, a study conducted by Shambrook [28] comprehensively surveyed Research Managers and Administrators (RMAs) from institutions such as the US, Canada (CARA), Europe (EARMA), Australia (ARMS), BESTPRAC, and the UK (ARMA) between 2007 and 2020. Approximately 46.6% of participants reported experiencing “high” or “extremely high” levels of stress, and 87.8% reported that their workload had increased significantly over the past few years.

The findings reveal that RMAs are considered one of the most stressful occupations globally.

One study emphasizing the need to assess stress not only at the individual level but also at the organizational and structural level is the compilation by Sonnentag et al. [29]. This study systematically examined the psychological factors that affect employee health and well-being, emphasizing that not only stressors, but also multidimensional factors, such as job resources, leadership style, individual behaviors, organizational interventions, and historical–cultural context, are determinants of job stress. This approach demonstrates that occupational stress cannot be reduced to a single source and that multifactorial assessments are necessary. Similarly, studies conducted in different geographical and institutional contexts reveal the dimensions and effects of occupational stress. Ablanedo-Rosas et al. [30] noted, in a study conducted among academic and administrative staff, as well as students, at a university in the USA, that the effects of stress differed across groups. Nevertheless, there were similarities in terms of age and gender. Students were found to be more affected by the negative consequences of stress. A study conducted by Ankomah and Dzikunu [31] at the University of Education, Winneba (UEW), in Ghana, examined the effects of occupational stress experienced by administrative staff on job satisfaction and health within the framework of the demand–control theory. The study found that high levels of perceived stress have a significantly negative impact on job satisfaction.

One of the studies evaluated explicitly within the academic community, Shen and Slater’s [32] integrative review, revealed that a large portion of academic staff experience moderate to high levels of stress. While workload stands out as the most significant source of stress, occupational stress has been reported to be associated with adverse outcomes such as depression, anxiety, burnout, and a decline in general well-being. These findings demonstrate that occupational stress is not limited to work but is also a serious risk factor that threatens the mental and physical health of academics. Similarly, Ghareeb Mohamed and Khamis Mohamed [33], in a study conducted at Hafr al Batin University in Saudi Arabia, reported that academics experienced moderate levels of occupational stress, which they assessed as “a problem but not acute.” Participants were found to employ individual strategies, such as walking, prayer, and positive thinking, to cope with stress. Hammoudi Halat et al. [34], a study conducted for the first time at universities in Qatar, examined the relationships between academic staff’s perception of occupational stress and symptoms of depression, anxiety, and stress. They found that 26% of participants experienced at least moderate levels of stress, and that workload-related stressors such as time pressure and lack of recognition were directly related to psychological symptoms. Samrah [35], in a study conducted at universities in the United Arab Emirates, examined the impact of perceived occupational stress on individual performance and organizational success among academic staff, showing that performance decreased as stress levels increased. In this context, personal perceptions, the work environment, and individual personality traits were highlighted as sources of stress. Another critical study focusing on the impact of stress on job satisfaction in the academic context was conducted by Jahanzeb [36] at a distance learning institution in Pakistan. Factors such as role ambiguity, role conflict, office politics, meaningless tasks, and administrative pressures were reported to have a significant negative impact on job satisfaction among academic staff. In this study, the overall median score of the Work Stress Scale was determined to be 2.22 ± 0.90. Similarly, in the study conducted by Teleş, the median score of the same scale was reported as 2.55 ± 0.87. A score closer to 1 on the scale indicates a low level of work-related stress, while a score closer to 5 indicates a high level of stress [3]. Accordingly, the findings of our study suggest that participants experienced a moderate level of work stress.

In addition to all these studies, the research conducted by Sapawi and Ming [37] at University Malaysia Sabah is noteworthy for demonstrating that occupational stress not only has psychological but also behavioral consequences. In the study, 58.8% of office workers reported experiencing moderate stress, while 35.9% reported experiencing high levels of stress. Female employees were found to have higher stress levels than their male counterparts, and stress was found to lead to an increase in the consumption of “comfort food”, particularly. This finding suggests that occupational stress can affect not only individuals’ mental state but also their daily life practices and health-related behaviors. In the study, female academics were found to experience higher levels of work-related stress than their male counterparts. When the relationship between work stress and eating habits was examined, the total median score of AEBQ-TR was determined as 19.26 ± 2.29, and it was also observed that female academics had higher scores in emotional over-eating, slowness in eating, and hunger + food responsiveness. This situation shows that women tend to eat more emotionally under stress, are more sensitive to hunger, and eat more slowly. On the other hand, emotional under-eating and food fussiness behaviors were observed to be more pronounced in male academicians. These results suggest that men’s stress coping mechanisms are different from women’s and that they tend to eat less under stress. It was determined that gender has a significant effect on eating behaviors: female academicians tend to eat emotionally under stress, while male academicians tend to avoid certain foods by changing their food preferences. Gonçalves et al. [38] revealed in their study on obese women (BMI > 30 kg/m^2^) that stress is mainly associated with emotional eating and external eating behaviors and that stress may have an effect on body fat distribution. This suggests that stress management may play an essential role in nutritional interventions.

In Türkiye, academic positions in state universities are organized in different statuses. While professors and associate professors serve as permanent civil servants, assistant professors and lecturers are subject to a contract renewal process every two years. While those in the associate professor position are required to produce academic publications by the appointment criteria determined by the university, lecturers are only obliged to teach. Academicians who have completed their doctorates must meet the appointment conditions to be promoted to the assistant professor position. Research assistants work in institutes with varying statuses, especially in public universities. If they are not appointed to an assistant professor position after completing their master’s and doctoral education, their relationship with the institution typically ends [39,40,41]. For these reasons, promotions in the academy are one of the critical factors that can cause stress on academicians. The study determined that professors had the lowest level of work stress, while the research assistants’ doctor and research assistants had the highest level of stress. This indicates that work stress decreases significantly as the title increases. In addition, professors had the lowest adult eating behavior scores, while research assistants and research assistants had the highest scores. This result supports the tendency to have more regular and healthy eating habits with increased academic titles.

### Strengths, Limitations, and Future Research

This study is the first to investigate the relationship between work stress and eating behaviors among Turkish academicians, utilizing valid and reliable scales. This study is valuable as it investigates the relationship between occupational stress and eating behaviors among academicians, a professional group facing increasing mental and physical demands. The study provides important preliminary findings for developing targeted strategies to support well-being, healthy eating, and stress management within academic institutions.

However, the study has several limitations. First, the use of snowball sampling may have introduced sampling bias, limiting the representativeness of the findings. Future studies should consider employing stratified random sampling to ensure a balanced representation across academic disciplines, institutional types (e.g., public vs. private), and geographic regions. Second, the cross-sectional nature of the study restricts causal interpretations. To better capture the temporal dynamics and identify the causal pathways, future research should adopt longitudinal designs or mixed methods approaches, integrating qualitative data such as interviews to explore underlying mechanisms. Moreover, although demographic variables such as age and gender were included, further details—such as field of study, institutional structure, and regional background—should be reported in order to enhance the contextual interpretation of the results. Finally, the practical implications of the study should be considered, considering institutional interventions. Based on the findings, recommendations could include university-based stress-reduction programs, nutrition education initiatives, and gender-sensitive policies that address the specific stressors and behavioral tendencies of different academic groups. In addition, incorporating individual-level factors such as coping skills, sleep patterns, and psychological well-being into future models may deepen the understanding of stress-related eating behaviors. Comparative studies across different cultural and institutional settings are also recommended in order to improve the generalizability of results.

## 5. Conclusions

This study examines the relationship between gender and work stress and eating behaviors. Female academicians experience more work stress and score higher on the eating behaviors measured on the AEBQ compared to male academicians. This suggests that female academicians may be more prone to eating behaviors related to emotional or environmental influences when coping with work stress. It was also observed that stress levels increased in individuals with lower academic titles; this may be due to the academic promotion processes and structural difficulties in Türkiye.

This study demonstrates the relationship between work stress experienced by academicians and eating behaviors, and how this relationship differs by gender. While numerous studies have examined the relationship between stress and eating behaviors, this is the first to address the relationship between work stress and eating habits among academicians in Türkiye. In this context, stress management and promoting healthy eating habits can play a critical role in fostering a healthier lifestyle, particularly among academics.

However, more in-depth research is needed to understand the role of eating styles, psychological factors, individual differences, and the social environment in this process. Future studies can contribute to the development of individualized intervention programs by examining how stress management strategies shape eating habits.

## Figures and Tables

**Figure 1 healthcare-13-01758-f001:**
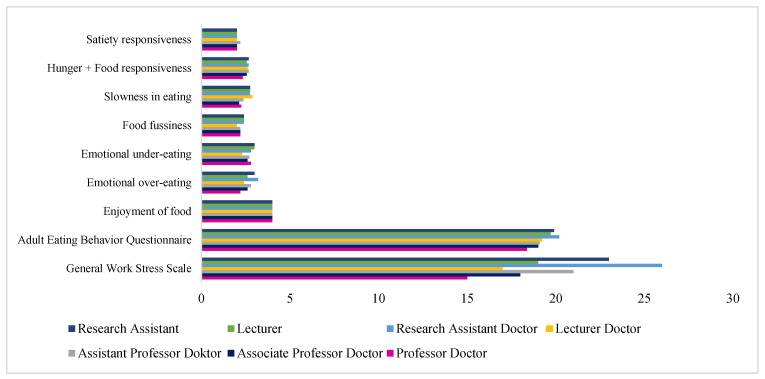
Median values of academicians eating behaviors and work stress by academic title.

**Table 1 healthcare-13-01758-t001:** Sociodemographic and health-related variables of the academicians (*n* = 649).

Variable	Categories	*n* (649)	% (100)
Gender	Female	397	61.2
	Male	252	38.8
Academic title	Professor Doctor	78	12.1
	Associate Professor Doctor	118	18.2
	Assistant Professor Doctor	172	26.5
	Lecturer Doctor	46	7.1
	Research Assistant Doctor	33	5.1
	Lecturer	101	15.6
	Research Assistant	101	15.6
What percentage of your monthly income is spent on food	0–25%	261	40.2
	25–50%	290	44.7
	50–75%	80	12.3
	75–100%	18	2.8
Diagnosed Disease	None	421	64.8
	Diabetes	25	3.8
	Cardiovascular diseases	29	4.5
	Rheumatic diseases	16	2.5
	Inflammatory bowel disease	6	0.9
	Gastritis-ulcer	38	5.9
	Other	114	17.6
Special diet for disease	None	480	74.0
	Weight Loss	77	11.9
	Low fat, cholesterol	21	3.2
	Salt-free, sodium-limited	10	1.5
	Diabetic compatible	24	3.7
	High fiber	7	1.1
	Other	30	4.6
Adequate and balanced nutritional status	Yes	313	48.2
	No	171	26.3
	I’m not sure	165	25.5

**Table 2 healthcare-13-01758-t002:** The academicians’ variables related to eating habits.

Variable	Categories	*n* (649)	% (100)
Main meal consumption status	1	38	5.9
	2	415	63.9
	3	196	30.2
Skipping main meal	None	111	17.1
	Breakfast	149	23.0
	Lunch	141	21.7
	Dinner	3	0.5
	Sometimes, skipping any meal	245	37.7
Reason for skipping meals		*n* (538)	% (100)
	Timelessness	256	47.6
	Living alone	31	5.8
	To lose weight	65	12.1
	Lack of appetite	183	34.0
	Economic inadequacy	3	0.5
Snacks meal consumption status	1	241	37.1
	2	154	23.7
	3	44	6.8
	None	210	32.4
Skipping snacks		*n* (439)	% (100)
	Mid-morning snacks	28	6.4
	Afternoon snacks	24	5.5
	Night-time snacks	25	5.7
	Sometimes, skipping any snacks	362	82.4
Reason for skipping snacks		*n* (439)	% (100)
	Timelessness	176	40.1
	I’m not used to it	114	26.0
	Living alone	31	7.1
	Lack of appetite	117	26.6
	Economic inadequacy	1	0.2

**Table 3 healthcare-13-01758-t003:** Eating behaviors and general work stress results of academicians according to their gender.

	Median		U	*p*-Value
	Female	Male		
GWSS	21.00	17.00	39,856.5	<0.001 *
AEBQ-TR	19.78	18.55	35,463.5	<0.001 *
Enjoyment of food	4.00	4.00	49,538.5	0.834
Emotional over-eating	3.00	2.20	38,840.0	<0.001 *
Emotional under-eating	2.60	2.80	49,429.0	0.798
Food fussiness	2.20	2.40	44,486.0	0.017 *
Slowness in eating	2.50	2.25	41,619.5	<0.001 *
Hunger + food responsiveness	2.67	2.44	42,349.0	0.001 *
Satiety responsiveness	2.20	2.00	37,382.5	<0.001 *

U: Mann–Whitney U Test; * *p* < 0.05; GWSS: General Work Stress Scale; AEBQ-TR: The Adult Eating Behavior Questionnaire.

**Table 4 healthcare-13-01758-t004:** Average scores of academicians’ work stress, eating behaviors, and their subscales, and the differences between them.

	Median							H	*p*-Value	Difference
	Prof. Dr.	Assoc. Prof. Dr.	Asst. Prof. Dr.	Lect. Dr.	Res. Asst. Dr.	Lect.	Res. Asst.			
GWSS	15.00	18.00	21.00	17.00	26.00	19.00	23.00	50.562	<0.001 *	I < II, III, IV, V, VII II < V, VII III < IV
AEBQ-TR	18.37	19.01	19.09	19.23	20.20	19.71	19.91	25.656	<0.001 *	I < V, VI, VII II < VII
Enjoyment of food	4.00	4.00	4.00	4.00	4.00	4.00	4.00	4.683	0.585	
Emotional over-eating	2.20	2.60	2.80	2.40	3.20	2.60	3.00	5.539	0.477	
Emotional under-eating	2.80	2.60	2.70	2.30	2.80	3.00	3.00	2.604	0.857	
Food fussiness	2.20	2.20	2.20	2.00	2.40	2.40	2.40	4.447	0.616	
Slowness in eating	2.25	2.13	2.38	2.88	2.75	2.75	2.75	10.079	0.121	
Hunger + food responsiveness	2.33	2.56	2.67	2.61	2.67	2.56	2.67	11.335	0.079	
Satiety responsiveness	2.00	2.00	2.20	2.00	2.00	2.00	2.00	2.421	0.877	

H: Kruskal–Wallis H test; * *p* < 0, 05; GWSS: General Work Stress Scale; AEBQ-TR: The Adult Eating Behavior Questionnaire; Prof. Dr.—I: Professor Doctor; Assoc. Prof. Dr.—II: Associate Professor Doctor; Asst. Prof. Dr.—III: Assistant Professor Doctor; Lect. Dr.—IV: Lecturer Doctor; Res. Asst. Dr—V: Research Assistant Doctor; Lect.—VI: Lecturer; Res. Asst.—VII: Research Assistant.

**Table 5 healthcare-13-01758-t005:** Correlation between GWSS and AEBQ-TR.

	GWSS	AEBQ-TR	Enjoyment of Food	Emotional Over-Eating	Emotional Under-Eating	Food Fussiness	Slowness in Eating	Hunger + Food Responsiveness	Satiety Responsiveness
GWSS	1								
AEBQ-TR	0.214 **	1							
Enjoyment of food	0.045	0.296 **	1						
Emotional over-eating	0.272 **	0.322 **	0.351 **	1					
Emotional under-eating	−0.054	0.225 **	−0.225 **	−0.575 **	1				
Food fussiness	0.061	0.260 **	−0.262 **	−0.012	0.037	1			
Slowness in eating	−0.064	0.506 **	−0.179 **	−0.199 **	0.207 **	0.011	1		
Hunger + food responsiveness	0.301 **	0.498 **	0.528 **	0.542 **	−0.203 **	−0.054	−0.146 **	1	
Satiety responsiveness	−0.032	0.294 **	−0.350 **	−0.273 **	0.391 **	0.073	0.351 **	−0.192 **	1

Sperman Correlation Test ** *p* < 0.01, GWSS: General Work Stress Scale; AEBQ-TR: The Adult Eating Behavior Questionnaire.

## Data Availability

The original contributions presented in this study are included in the article. Further inquiries can be directed to the corresponding author.

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
