# Peer review of "Effect of Work Stress and Eating Behavior: A Study Among Academicians in Türkiye"

_healthcare, 2025, doi:10.3390/healthcare13141758_

Round 1
Reviewer 1 Report
Comments and Suggestions for Authors
This study is an important research examining the relationship between work stress and eating behaviors among academicians in Turkey, focusing on gender and academic title. Using a large sample and reliable measurement tools, the findings reveal gender-specific effects of stress on eating habits and the role of academic position. The study provides valuable data for stress management and promoting healthy eating habits within the academic environment.
My general opinion about the article is that although the methods, results, and discussion sections are well written, the introduction needs improvement. My suggestions are as follows:
-
The abstract could include that the study is cross-sectional and specify the data collection period.
-
The introduction text is somewhat fragmented in terms of topic coherence. Instead of focusing on the relationship between academic stress and eating behavior, the first two paragraphs are quite general and lengthy. Topics such as the general functions of universities and the etiology of stress should be summarized more briefly. The text should be addressed directly in the context of academic workload, stress, and its effect on nutrition.
-
The information in lines 73-76: “It has been observed that 40-50% of individuals under stress increase their food intake, and at the same time, stress causes an increase in unhealthy food consumption [8].” should be clarified more explicitly.
-
A literature gap should be clearly stated, such as “This issue has not been sufficiently studied specifically among academicians in Turkey.” Or if there are studies on this subject, they should be mentioned.
-
It is mentioned that the survey was conducted online, but the platform used (Google Forms, Qualtrics?) and distribution channels (email, social media) are not specified. The data collection tool and distribution process should be described more clearly.
-
In the discussion, the paragraph starting with “In Türkiye, academic positions in state universities are organized in different statuses. While professors and associate professors serve as permanent civil servants, assistant professors and lecturers are subject to a contract renewal process every two years. While those in the associate professor position are required to produce academic…” should be supported with references.
Author Response
Please see the attachment.
Comments 1: The abstract could include that the study is cross-sectional and specify the data collection period.
Response 1: Cross-sectional and study data collection period information was added to the abstract based on the reviewer's suggestion.
Comments 2: The introduction text is somewhat fragmented in terms of topic coherence. Instead of focusing on the relationship between academic stress and eating behavior, the first two paragraphs are quite general and lengthy. Topics such as the general functions of universities and the etiology of stress should be summarized more briefly. The text should be addressed directly in the context of academic workload, stress, and its effect on nutrition.
Response 2: In line with the reviwer’s suggestion, the introduction section has been revised to incorporate more specific information."
Comments 3: The information in lines 73-76: “It has been observed that 40-50% of individuals under stress increase their food intake, and at the same time, stress causes an increase in unhealthy food consumption [8].” should be clarified more explicitly.
Response 3: We have reviewed the statement in lines 73–76, "It has been observed that 40-50% of individuals under stress increase their food intake, and at the same time, stress causes an increase in unhealthy food consumption [8]." and revised it to more accurately reflect the findings of the cited meta-analysis. Instead of emphasizing a specific percentage of individuals, we now highlight the types of foods consumed under stress, particularly the preference for highly palatable and energy-dense foods.
Comments 4: A literature gap should be clearly stated, such as “This issue has not been sufficiently studied specifically among academicians in Turkey.” Or if there are studies on this subject, they should be mentioned.
Response 4: This information is presented under the section titled Strengths, Limitations, and Future Research.
Comments 5: It is mentioned that the survey was conducted online, but the platform used (Google Forms, Qualtrics?) and distribution channels (email, social media) are not specified. The data collection tool and distribution process should be described more clearly.
Response 5: “The platform where the survey was conducted online is Google Form, and the distribution channel is the Electronic Document Management System (EDMS) used in private and public institutions. EDMS is an integrated software system that allows the creation, signing, transmission and archiving of internal and external correspondence in a digital environment.” This information has been added to the article.
Comments 6: In the discussion, the paragraph starting with “In Türkiye, academic positions in state universities are organized in different statuses. While professors and associate professors serve as permanent civil servants, assistant professors and lecturers are subject to a contract renewal process every two years. While those in the associate professor position are required to produce academic…” should be supported with references.
Response 6: A proper reference has been provided to support this information.

Reviewer 2 Report
Comments and Suggestions for Authors
Dear Author
This is a well-designed study with significant contributions to understanding stress and eating behaviors in academia. Addressing the limitations in future research would further strengthen its impact. The manuscript is suitable for publication.
There are some recommendations for the future studies.
- Address Sampling Bias : Future studies could use stratified random sampling to ensure representation across institutions, regions, and disciplines.
2. Longitudinal or Mixed-Methods Approach: Consider tracking stress and eating behaviors over time or supplementing surveys with interviews to explore underlying mechanisms.
3. Expand Demographic Reporting: Include details about participants’ disciplines, institutional types (public/private), and regional distribution to enhance contextual understanding.
4. Clarify Practical Implications**: Provide specific recommendations for universities (e.g., stress-reduction programs, gender-sensitive interventions) based on the findings.
Regards,
Reviewer
Author Response
Please see the attachment.
Comments 1: There are some recommendations for the future studies.
- Address Sampling Bias : Future studies could use stratified random sampling to ensure representation across institutions, regions, and disciplines.
- Longitudinal or Mixed-Methods Approach: Consider tracking stress and eating behaviors over time or supplementing surveys with interviews to explore underlying mechanisms.
- Expand Demographic Reporting: Include details about participants’ disciplines, institutional types (public/private), and regional distribution to enhance contextual understanding.
- Clarify Practical Implications**: Provide specific recommendations for universities (e.g., stress-reduction programs, gender-sensitive interventions) based on the findings.
Response 1: Based on the reviwer’s feedback, the 'Strengths, Limitations, and Future Research' section has been revised accordingly.

Reviewer 3 Report
Comments and Suggestions for Authors
Review of the manuscript titled: Effect of Work Stress on Eating Behaviour: A Study among Academicians in Türkiye
Overall, I find the manuscript interesting and relevant. Investigating the causes of eating behaviour is a valuable endeavour, as it may contribute to the prevention of serious health issues such as obesity or eating disorders, particularly those involving binge eating or disordered eating patterns. The manuscript touches upon a socially significant topic and offers insight into how occupational stress may influence dietary habits, especially in academic settings. However, although the topic is meaningful, and the study has potential, to meet the standards of scientific rigour and clarity expected by the journal, the authors should consider revising the manuscript with greater attention to methodological transparency and the theoretical framework supporting the research.
Below, I provide detailed comments and suggestions, organised by sections of the manuscript.
Keywords: I recommend replacing the keyword “stress” with “occupational stress” to better reflect the focus of the study. Different types of stress can have varying impacts on eating behaviour, so it would be more accurate and specific to the research.
Introduction:
- The introduction would benefit from a more comprehensive explanation of stress in general, including its aetiology, prevalence, distinctions between acute and chronic stress, and the primary consequences of different types of stress. After presenting a general background, a focused explanation of "occupational stress" should be introduced. This would involve outlining its unique features and specific causes, which differ from those of other types of stress. These additions would strengthen the rationale for the study.
- Currently, there is insufficient contextual information and literature to justify the research question and objectives. A clearer exposition of why this particular study is necessary and valuable is needed.
- Regarding the stated aims of the study: the sentence “showing the effects of both individual awareness and work life” is unclear. As far as I can tell, individual awareness does not appear to be explicitly measured or analysed in the study. If I am mistaken, this could be clarified; if not, this objective should be rephrased or omitted for clarity and accuracy.
Materials and Methods:
- Study Design: Some of the content presented in this section does not align with what is typically included under this heading. It would be helpful to reorganise the structure of the Methods section for clarity.
- Participants, Recruitment & Sample: a) The term “participants” appears redundant with “sample” — please clarify and unify terminology as appropriate; b) The description of the sample is insufficient. Only the statistical sample size calculation is discussed, which, although important, is not sufficient in place of a detailed description of the actual sample used. Please specify: The total number of participants, Distribution by gender and age, Recruitment procedures, Inclusion/exclusion criteria…
- Data Collection: This section currently focuses on the assessment tools rather than on the data collection process itself. The assessment tools described here would be more appropriately discussed in a separate section titled “Instruments” or “Assessment Tools”. I recommend indicating in this section: How participants were contacted and how data were collected (e.g. online, in person), Who collected the data and under what conditions, Time required to complete the assessments, Period of data collection, Informed consent procedures, Measures taken to ensure anonymity and confidentiality...
- Instruments (or Scales / Assessment Tools): As previously said, a separate section should be devoted to the tools used for data collection. Along with the already-mentioned GWSS and AEBQ, the socio-demographic questionnaire designed ad hoc for this study should also be described briefly. Regarding the AEBQ: Please state the number of items included in the questionnaire. Consider including an appendix with the items or sample questions for better understanding, particularly for readers unfamiliar with the instrument (same with the GWSS). The distinction between the four “food approach” and three “food avoidance” sub-dimensions should be presented more clearly (line 134). A short list or table may enhance clarity and comprehension.
Results:
The reliability data provided in lines 163 to 171 regarding the psychometric properties of the instruments used (e.g., Cronbach's alpha values) are informative. However, this information would be more appropriately placed in the Instruments section rather than in Results. Since the primary aim of the study is not to assess the psychometric properties of the scales but rather to investigate the relationship between occupational stress and eating behaviour, reporting reliability here seems somewhat out of place.
Several tables in this section require clarification and improvement: Percentages do not always sum to 100% — this should be verified and corrected. In Table 1, the item “What percent of your monthly expenses income?” is unclear. It should be rephrased to clarify whether it refers to the percentage of income spent on food or some other category. Percentage signs (%) should follow the numerical values (e.g., “0–25%”), not precede them (“%0–25”), in accordance with English-language conventions.
There is redundancy between the narrative text and some tables; rather than repeating the numbers, the text should highlight key trends or unexpected findings. For example, when discussing Table 3, it would be useful to briefly explain what higher scores in specific subscales such as “Food Fussiness” or “Food Responsiveness” indicate in behavioural terms.
Several variables presented appear disconnected from the aims of the study. For instance, BMI values are reported, yet they are not incorporated into any statistical analysis. If BMI was included to demonstrate that the sample is representative of the national population, this should be explicitly stated — ideally supported by national data. As it stands, the fact that the sample's mean BMI falls into the overweight range, with approximately half of participants classified as pre-obese or obese, raises the question: is this reflective of the broader Turkish academic population? If not, this may represent a sampling bias that should be acknowledged and discussed. Similarly, other variables such as “income” and “daily water consumption” are presented in descriptive statistics but are not subsequently employed in any analytical procedures. Their relevance to the research objectives is unclear, and unless justified, they should either be removed or integrated into the analyses.
The sentence “the average eating behaviour scores of male are lower than those of female” is ambiguous. It is not clear whether this means that women eat “more,” “worse,” or differently. Since the meaning of higher or lower total scores on the AEBQ has not been explained in the manuscript, the significance of this result is difficult to interpret. A brief description of what high or low scores represent behaviourally would be helpful.
The manuscript states that “Spearman correlation analysis was conducted to assess the effect of work stress on eating behaviours” (line 257). However, a correlation analysis does not establish causal relationships — only associations. Therefore, the phrase “to assess the effect” should be revised.
Regarding the strength of the correlations: the manuscript claims the relationships are “low” (line 268) or “moderate” (line 269). However, no justification is provided for these classifications. The thresholds or reference criteria used (e.g. Cohen’s guidelines) should be cited to support these labels.
The chi-square test is used to analyse associations between categorical variables. However, the GWSS and AEBQ are not inherently categorical. The manuscript does not specify how the continuous total scores from these instruments were categorised (e.g., low/medium/high) to allow for chi-square analysis. This procedure needs to be explained clearly.
The central focus of the study — the relationship between work stress and eating behaviours — is only minimally addressed in the results, most notably in Table 5. Instead, the manuscript devotes considerable attention to group differences by gender and academic title, which appear secondary to the main objective. If socio-demographic variables are believed to moderate or mediate the relationship between stress and eating behaviours, appropriate interaction or regression analyses should be conducted and reported.
The inclusion of BMI in the “scales” section suggests it was considered a relevant variable, yet it is never used analytically. It would have been insightful to explore how BMI relates to occupational stress and/or eating behaviour.
Subheadings within the Results section (e.g., “Gender Differences,” “Stress-Eating Behaviour Relationship”) would aid structure and readability.
Finally, Figure 1 appears redundant and does not offer added value beyond the corresponding table. It may be omitted to improve conciseness.
Discussion:
The initial paragraph of the Discussion provides theoretical background on stress and eating behaviour, which would have been more appropriately placed in the Introduction as part of the literature review. Similarly, references to stress-related eating in children are not relevant here, given that the study population consists of adults.
The section focuses disproportionately on gender differences and academic title without sufficiently discussing the relationship between occupational stress and eating behaviour, which was the stated primary aim. This is a significant omission. The results do not explore in depth how eating behaviour changes with different levels of occupational stress, nor are these findings contextualised within the existing literature. The authors assert (e.g., in the Strengths section) that “this study is the first to investigate the relationship between job stress and eating behaviours among Turkish academicians.” If this is indeed the primary contribution, the discussion should centre on this point, drawing comparisons with similar studies, considering cultural/contextual implications, and offering possible interpretations for the observed patterns. If instead the true focus of the study was to examine gender- or title-based differences in either stress or eating behaviour, then the framing of the entire paper — title, introduction, objectives — should be revised to reflect this.
Claims about “strong statistical analyses” in the Strengths section should be moderated or substantiated. As noted earlier, certain statistical methods (e.g., chi-square tests, correlation strength interpretation) are not well justified, and key variables are under-analysed.
Conclusion:
The opening statement of the Conclusion — “this study reveals the effects of stress on eating habits” — is not supported by the presented data. Correlation analyses do not permit causal inference, and no advanced modelling (e.g., regression, path analysis) was used to explore the “effect” of one variable on another.
The conclusion should be revised to reflect the actual findings of the study, which primarily consist of observed associations (of varying strength) between work stress and certain dimensions of eating behaviour. If gender and academic title were important differentiators, that may also be acknowledged — but within the limitations of the design.
To sum up: The manuscript should either refocus the paper to centre on gender and academic status differences in stress and eating behaviour, updating the title, objectives, and framing accordingly; or revise and extend the statistical analyses to address the original aim: exploring the relationship between work stress and eating behaviour, possibly moderated by socio-demographic variables such as gender, academic title or BMI.
Author Response
Please see the attachment.
Keywords:
Comments 1: I recommend replacing the keyword “stress” with “occupational stress” to better reflect the focus of the study. Different types of stress can have varying impacts on eating behaviour, so it would be more accurate and specific to the research.
Response 1: According to your suggestion, the keyword "stress" has been changed to "occupational stress".
Introduction:
Comment 2: The introduction would benefit from a more comprehensive explanation of stress in general, including its aetiology, prevalence, distinctions between acute and chronic stress, and the primary consequences of different types of stress. After presenting a general background, a focused explanation of "occupational stress" should be introduced. This would involve outlining its unique features and specific causes, which differ from those of other types of stress. These additions would strengthen the rationale for the study.
Response 2: Upon suggestion, information about stress and occupational stress was added to the introduction section.
Comment 3: Currently, there is insufficient contextual information and literature to justify the research question and objectives. A clearer exposition of why this study is necessary and valuable is needed.
Response 4: This study is valuable as it investigates the relationship between occupational stress and eating behaviors among academicians in Türkiye, a professional group facing increasing mental and physical demands. Academic roles involve varying responsibilities and job security depending on title, which may influence stress levels and lifestyle behaviors. By examining how work-related stress is associated with changes in eating patterns—especially in relation to gender and academic status—the study offers important insights for developing targeted strategies to support well-being, healthy eating, and stress management within academic institutions. And also this study is considered important as it is the first to examine the eating behaviors of academicians within a sample from Türkiye.
Comment 4: Regarding the stated aims of the study: the sentence “showing the effects of both individual awareness and work life” is unclear. As far as I can tell, individual awareness does not appear to be explicitly measured or analysed in the study. If I am mistaken, this could be clarified; if not, this objective should be rephrased or omitted for clarity and accuracy.
Response 4: In the statement of the aim of the study, an incorrect and unclear expression was used, and it was revised clearly and explicitly and corrected in the text.
Materials and Methods:
Comment 5: Study Design: Some of the content presented in this section does not align with what is typically included under this heading. It would be helpful to reorganise the structure of the Methods section for clarity.
Response 5: The methods section structure has been rearranged.
Comment 6: Participants, Recruitment & Sample: a) The term “participants” appears redundant with “sample” — please clarify and unify terminology as appropriate; b) The description of the sample is insufficient. Only the statistical sample size calculation is discussed, which, although important, is not sufficient in place of a detailed description of the actual sample used. Please specify: The total number of participants, Distribution by gender and age, Recruitment procedures, Inclusion/exclusion criteria…
Response 6: The methods section structure has been rearranged.
Comment 7: Data Collection: This section currently focuses on the assessment tools rather than on the data collection process itself. The assessment tools described here would be more appropriately discussed in a separate section titled “Instruments” or “Assessment Tools”. I recommend indicating in this section: How participants were contacted and how data were collected (e.g. online, in person), Who collected the data and under what conditions, Time required to complete the assessments, Period of data collection, Informed consent procedures, Measures taken to ensure anonymity and confidentiality...
Response 7: The methods section structure has been rearranged.
Comment 8: Instruments (or Scales / Assessment Tools): As previously said, a separate section should be devoted to the tools used for data collection. Along with the already-mentioned GWSS and AEBQ, the socio-demographic questionnaire designed ad hoc for this study should also be described briefly. Regarding the AEBQ: Please state the number of items included in the questionnaire. Consider including an appendix with the items or sample questions for better understanding, particularly for readers unfamiliar with the instrument (same with the GWSS). The distinction between the four “food approach” and three “food avoidance” sub-dimensions should be presented more clearly (line 134). A short list or table may enhance clarity and comprehension.
Response 8: The evaluation tools section has been revised and written.
Results:
Comment 9: The reliability data provided in lines 163 to 171 regarding the psychometric properties of the instruments used (e.g., Cronbach's alpha values) are informative. However, this information would be more appropriately placed in the Instruments section rather than in Results. Since the primary aim of the study is not to assess the psychometric properties of the scales but rather to investigate the relationship between occupational stress and eating behaviour, reporting reliability here seems somewhat out of place.
Response 9:In accordance with the referee's suggestion, the first paragraph in the "Results" section has been moved to the "Methods" section.
Comment 10: Several tables in this section require clarification and improvement: Percentages do not always sum to 100% — this should be verified and corrected. In Table 1, the item “What percent of your monthly expenses income?” is unclear. It should be rephrased to clarify whether it refers to the percentage of income spent on food or some other category. Percentage signs (%) should follow the numerical values (e.g., “0–25%”), not precede them (“%0–25”), in accordance with English-language conventions.
Response 10: The tables have been revised; percentage signs (%) and numerical values (e.g., 0–25%) were corrected to reflect values out of 100%. In Table 1, the phrase “What percent of your monthly food expenses income?” has been replaced with the clearer version: “What percentage of your monthly income is spent on food?
Comment 11: There is redundancy between the narrative text and some tables; rather than repeating the numbers, the text should highlight key trends or unexpected findings. For example, when discussing Table 3, it would be useful to briefly explain what higher scores in specific subscales such as “Food Fussiness” or “Food Responsiveness” indicate in behavioural terms.
Response 11: Additional explanations have been added to the table upon your suggestion.
Comment 12: Several variables presented appear disconnected from the aims of the study. For instance, BMI values are reported, yet they are not incorporated into any statistical analysis. If BMI was included to demonstrate that the sample is representative of the national population, this should be explicitly stated — ideally supported by national data. As it stands, the fact that the sample's mean BMI falls into the overweight range, with approximately half of participants classified as pre-obese or obese, raises the question: is this reflective of the broader Turkish academic population? If not, this may represent a sampling bias that should be acknowledged and discussed. Similarly, other variables such as “income” and “daily water consumption” are presented in descriptive statistics but are not subsequently employed in any analytical procedures. Their relevance to the research objectives is unclear, and unless justified, they should either be removed or integrated into the analyses.
Response 12: Analysis results that were outside our study objective were removed from the text and table upon your suggestion.
Comment 13: The sentence “the average eating behaviour scores of male are lower than those of female” is ambiguous. It is not clear whether this means that women eat “more,” “worse,” or differently. Since the meaning of higher or lower total scores on the AEBQ has not been explained in the manuscript, the significance of this result is difficult to interpret. A brief description of what high or low scores represent behaviourally would be helpful.
Response 13: The interpretation of the AEBQ total score has been added to both Table 3 and the scale description section.
Comment 14: The manuscript states that “Spearman correlation analysis was conducted to assess the effect of work stress on eating behaviours” (line 257). However, a correlation analysis does not establish causal relationships — only associations. Therefore, the phrase “to assess the effect” should be revised.
Response 14: Upon your suggestion, the statement was changed to "The relationship between work stress and eating behaviors was evaluated using Spearman correlation analysis."
Comment 15: Regarding the strength of the correlations: the manuscript claims the relationships are “low” (line 268) or “moderate” (line 269). However, no justification is provided for these classifications. The thresholds or reference criteria used (e.g. Cohen’s guidelines) should be cited to support these labels.
Response 15: As Table 5 presents the results of the Spearman correlation analysis, relevant information regarding the interpretation of Spearman correlations has been included in the preceding paragraph
Comment 16: The chi-square test is used to analyse associations between categorical variables. However, the GWSS and AEBQ are not inherently categorical. The manuscript does not specify how the continuous total scores from these instruments were categorised (e.g., low/medium/high) to allow for chi-square analysis. This procedure needs to be explained clearly.
Response 16: The chi-square test was used solely within the scope of the post-hoc power analysis to assess whether the sample size was statistically sufficient. Other than this purpose, chi-square test analysis was not applied in the study.
Comment 17: The central focus of the study — the relationship between work stress and eating behaviours — is only minimally addressed in the results, most notably in Table 5. Instead, the manuscript devotes considerable attention to group differences by gender and academic title, which appear secondary to the main objective. If socio-demographic variables are believed to moderate or mediate the relationship between stress and eating behaviours, appropriate interaction or regression analyses should be conducted and reported.
Response 17: You suggested conducting interaction or regression analyses to examine the moderating or mediating effects of socio-demographic variables on the relationship between work stress and eating behaviors. However, since the sample distribution among subgroups such as academic title and gender did not provide sufficient statistical power for such analyses, the study focused on descriptive and correlational findings in line with its initially determined purpose. The findings offer significant contributions to the relationship between work stress and eating behaviors in academicians.
Comment 18: The inclusion of BMI in the “scales” section suggests it was considered a relevant variable, yet it is never used analytically. It would have been insightful to explore how BMI relates to occupational stress and/or eating behaviour.
Response 18: BMI classification was removed from the study and method.
Comment 19: Subheadings within the Results section (e.g., “Gender Differences,” “Stress-Eating Behaviour Relationship”) would aid structure and readability.
Response 19: Subheadings were added to the results section upon your suggestion.
Comment 20: Finally, Figure 1 appears redundant and does not offer added value beyond the corresponding table. It may be omitted to improve conciseness.
Response 20: Figure 1 is omitted from the article.
Discussion:
Comment 21: The initial paragraph of the Discussion provides theoretical background on stress and eating behaviour, which would have been more appropriately placed in the Introduction as part of the literature review. Similarly, references to stress-related eating in children are not relevant here, given that the study population consists of adults.
Response 21: In line with your suggestion, the first paragraph of the discussion section has been moved to the introduction section. Information on children has been removed.
Comment 22: The section focuses disproportionately on gender differences and academic title without sufficiently discussing the relationship between occupational stress and eating behaviour, which was the stated primary aim. This is a significant omission. The results do not explore in depth how eating behaviour changes with different levels of occupational stress, nor are these findings contextualised within the existing literature. The authors assert (e.g., in the Strengths section) that “this study is the first to investigate the relationship between job stress and eating behaviours among Turkish academicians.” If this is indeed the primary contribution, the discussion should centre on this point, drawing comparisons with similar studies, considering cultural/contextual implications, and offering possible interpretations for the observed patterns. If instead the true focus of the study was to examine gender- or title-based differences in either stress or eating behaviour, then the framing of the entire paper — title, introduction, objectives — should be revised to reflect this.
Response 22: In the context of Turkey, the structural problems related to the insecurity of tenure and promotion processes experienced by academics have been evaluated as an important source of occupational stress. Therefore, in order to reveal the effect of job stress on eating behaviors, which occurs in line with variables such as academic title and gender, these factors were particularly emphasized. In this context, explanatory information on the academic staff structure in Turkey was included in the discussion section and the purpose of the study was revised to better reflect this context. In addition, the discussion section was strengthened by supporting it with existing literature addressing the effect of stress on eating behaviors.
Comment 23: Claims about “strong statistical analyses” in the Strengths section should be moderated or substantiated. As noted earlier, certain statistical methods (e.g., chi-square tests, correlation strength interpretation) are not well justified, and key variables are under-analysed.
Response 23: The statements in the Strengths, Limitations, and Future Research section have been moderated to better reflect the scope and depth of the statistical methods used. References to “strong statistical analyses” have been revised to avoid overstating the analysis. We have also clarified the limitations of the applied tests (e.g., chi-square and Spearman correlation) and acknowledged the absence of multivariate analysis as a constraint in interpreting relationships.
Conclusion:
Comment 24: The opening statement of the Conclusion — “this study reveals the effects of stress on eating habits” — is not supported by the presented data. Correlation analyses do not permit causal inference, and no advanced modelling (e.g., regression, path analysis) was used to explore the “effect” of one variable on another.
Response 24: The first paragraph of the conclusion section was revised and written according to the purpose of the study.
Comment 25: The conclusion should be revised to reflect the actual findings of the study, which primarily consist of observed associations (of varying strength) between work stress and certain dimensions of eating behaviour. If gender and academic title were important differentiators, that may also be acknowledged — but within the limitations of the design.
Response 25: The conclusion section was revised and written according to the purpose of the study.
Reviewer 4 Report
Comments and Suggestions for Authors
Very clearly written work without unnecessary complications. It is a very interesting consideration.
The first paragraph in the "Results" section should be moved to the "Methods" section.
In front of reference 15, a dot is placed - move behind the reference
In Table 1, add the minimum and maximum for age and BMI.
In Table 2 - Skipping main meal and Skipping snacks does not add up to 100% due to decimal rounding.
It would be good to enter values for Figure 2, but if it is too compact, it can be omitted
“The study determined that professors had the lowest level of work stress, while research assistants and research assistants had the highest level of stress” – correct, research assistants was written twice
Author Response
Please see the attachment.
Comments 1: The first paragraph in the "Results" section should be moved to the "Methods" section.
Response 1: In accordance with the referee's suggestion, the first paragraph in the "Results" section has been moved to the "Methods" section.
Comments 2: In front of reference 15, a dot is placed - move behind the reference
Response 2: As per your suggestion, the dot before reference 15 has been relocated.
Comments 3: In Table 1, add the minimum and maximum for age and BMI.
Response 3: Table 1 includes the minimum and maximum values of age and BMI
Comments 4: In Table 2 - Skipping main meal and Skipping snacks does not add up to 100% due to decimal rounding.
Response 4: Table 2 - Skipping main meal and skipping snacks are recalculated to add 100%
Comments 5: It would be good to enter values for Figure 2, but if it is too compact, it can be omitted
Response 5: Although we appreciate your suggestion to include values in Figure 2, we refrained from doing so to prevent visual clutter. The corresponding numerical data are already presented in the main text.
Comments 6: “The study determined that professors had the lowest level of work stress, while research assistants and research assistants had the highest level of stress” – correct, research assistants was written twice
Response 6: The sentence has been revised and rewritten.
“The study determined that Professors had the lowest level of work stress, while research assistants’ doctor and research assistants had the highest level of stress.”

Round 2
Reviewer 3 Report
Comments and Suggestions for Authors
Although the authors have made some changes following the previous review, many of the issues previously raised remain unresolved. The manuscript still presents important shortcomings in its conceptual framework, methodological clarity, presentation of results, and the interpretation of findings. These aspects significantly limit the clarity, consistency, and scientific value of the study. Detailed comments are provided below.
Introduction:
The introduction often lacks a coherent narrative and instead reads as a sequence of disconnected data points. Additionally, certain ideas, such as the relationship between exposure to stress and eating behaviours, are unnecessarily repeated. In the previous round of revisions, the authors were explicitly asked to provide “a more comprehensive explanation of stress in general, including its aetiology, prevalence, distinctions between acute and chronic stress, and the primary consequences of different types of stress. After presenting a general background, a focused explanation of 'occupational stress' should be introduced." This request has not been adequately addressed. The discussion of stress remains superficial, and only a brief definition of "occupational stress" is provided. This is insufficient, particularly considering that the study is centred on the relationship between "occupational stress" and eating behaviors. Information on occupational stress should be developed in greater depth and placed in a separate, clearly defined paragraph.
In line 77, the use of “they” is unnecessary and should be replaced with “Hille et al. ... suggest...”.
Aims:
The aim is still stated as “to understand the effect of occupational stress on eating behaviours”. However, the study design and analyses presented do not allow for conclusions regarding causality. This issue was already noted in the previous review. The wording should be revised throughout the manuscript, including in the title, to reflect the descriptive and correlational nature of the study.
Materials and Methods:
- Study Design: The section should begin by explicitly stating the type of study conducted (e.g., observational, descriptive, cross-sectional), which is standard in this section.
- Participants: A specific subsection describing the participants is still missing. This section should include detailed information on sample characteristics and explain how anonymity and data confidentiality were ensured.
- Assessment Tools: This section requires restructuring and clearer writing. I recommend dividing it into three subsections:
1. Ad hoc socio-demographic questionnaire
2. German Workplace Stress Scale (GWSS)
3. Adult Eating Behaviour Questionnaire (AEBQ). The manuscript still does not clearly define what the four "food approach" and the three "food avoidance" subscales are, nor what they imply behaviourally. This clarification is essential.
An appendix should be included with sample items or a full list of questions, especially for the ad hoc questionnaire.
Within the descriptions of the standardised instruments, the authors should provide brief information on their psychometric properties (e.g., validity and reliability). This content should be incorporated into the subsections corresponding to each instrument.
- Reliability of Questionnaires: This section is unnecessary in its current form. It is not required to explain the general meaning of Cronbach’s alpha; instead, the authors should report the specific alpha values obtained for each instrument in the current study, and this information should be included in the relevant instrument subsections.
Results:
There remains a lack of explanation regarding what higher or lower scores in certain subscales—such as "Food Fussiness" or "Food Responsiveness"—indicate in behavioural terms. Although the authors claim to have added an interpretation to Table 3 and the scale description section, what has actually been included is a speculative interpretation of the results, which belongs in the conclusion, not in the results section. Without a proper explanation of the scale constructs and score meaning, the results are difficult to interpret.
The sentence “the average eating behaviour scores of male are lower than those of female” remains ambiguous. As noted previously, it is unclear whether this means women eat more, eat differently, or exhibit different behavioural patterns. Without a clear explanation of what high or low AEBQ scores indicate, the significance of this result remains unclear.
Moreover, the sentence “as the general work stress score increases, the AEBQ score increases” is not meaningful without specifying which dimensions of each scale are involved.
Regarding statistical analysis, the previous review noted: “The chi-square test is used to analyse associations between categorical variables. However, the GWSS and AEBQ are not inherently categorical. The manuscript does not specify how the continuous total scores from these instruments were categorised (e.g., low/medium/high) to allow for chi-square analysis.” The authors now claim that the chi-square test was only used in the context of the post-hoc power analysis. However, the information presented in Table 4 clearly includes chi-square comparisons across academic status groups. Thus, if categorical variables derived from continuous scores were used, the method for this categorisation must be explained. Otherwise, the table and corresponding text (lines 249–275) should be removed.
The authors state that “the study focused on descriptive and correlational findings in line with its initially determined purpose”. However, this is not reflected consistently across the manuscript, starting with the title and continuing through the aims, analyses, and conclusions.
Subsections were suggested for the Results section, but only two have been included, and neither aligns well with the main aim of the study (i.e., the relationship between occupational stress and eating behaviours).
There is still redundancy between the narrative and the tables. For example, "reasons to skip meals" appear both in Table 2 and in the main text. Similarly, Figure 1 and Table 3 seem to convey overlapping information.
Typographic conventions should be observed—for instance, the letter *n* should appear in italics
In Table 4, the title is misleading. What is presented are differences in certain variables according to academic status, not associations between variables.
In Table 1, the inclusion of age and BMI in columns labelled "n" and "%" is not appropriate. Besides, these values are already reported in the text, and BMI is not even used in the analyses, so these rows could be removed.
Discussion:
In this section, the authors should support their actual findings with studies specifically addressing the relationship between occupational stress and eating behaviour within academic settings, which is the stated focus of the study—or, at the very least, in other workplace environments. Studies involving adolescents or university students do not appear particularly relevant in this context. The study involving nurses might be more applicable, although stress in healthcare professions (e.g., among doctors and nurses) is likely not comparable to that experienced in academic settings.
In any case, the purpose of the discussion is not to present a list of unrelated studies—some more relevant than others—but rather to interpret and contextualise the results obtained, using the literature to shed light on the findings. The current discussion reads as the previous version with additional studies inserted, yet with no clear connection to the content that follows. As such, it lacks coherence and does not adequately engage with the study’s own results.
Conclusions:
The conclusions drawn are not supported by the results. The findings indicate: A low positive correlation between work stress and emotional overeating; A moderate correlation between work stress and hunger/food responsiveness; and no significant correlation between work stress and other eating behaviours (enjoyment of eating, emotional undereating, picky eating, or satiety sensitivity). Yet, the conclusion states: “This study reveals the effect of stress on eating habits and how this effect differs according to gender; it also shows that as academic title increases, significant changes are observed in stress levels and related eating habits scores.” This interpretation is problematic for several reasons:
- The word "effect" suggests causality, which the study design does not support.
- There does not even appear to be any statistically significant or substantively meaningful correlation between the variables."
- The moderating role of gender or academic status has not been tested — the study only examined whether there are group differences in stress or eating behaviour scores separately.
- The conclusion extrapolates beyond what the data support.
Given the number and persistence of the unresolved issues—many of which were already highlighted in the initial review—I do not consider the manuscript suitable for publication in its current form.
Author Response
Introduction:
Comments 1: The introduction often lacks a coherent narrative and instead reads as a sequence of disconnected data points. Additionally, certain ideas, such as the relationship between exposure to stress and eating behaviours, are unnecessarily repeated. In the previous round of revisions, the authors were explicitly asked to provide “a more comprehensive explanation of stress in general, including its aetiology, prevalence, distinctions between acute and chronic stress, and the primary consequences of different types of stress. After presenting a general background, a focused explanation of 'occupational stress' should be introduced." This request has not been adequately addressed. The discussion of stress remains superficial, and only a brief definition of "occupational stress" is provided. This is insufficient, particularly considering that the study is centred on the relationship between "occupational stress" and eating behaviors. Information on occupational stress should be developed in greater depth and placed in a separate, clearly defined paragraph.
Response 1: The introduction topic has been revised.
Aims:
Comments 2: The aim is still stated as “to understand the effect of occupational stress on eating behaviours”. However, the study design and analyses presented do not allow for conclusions regarding causality. This issue was already noted in the previous review. The wording should be revised throughout the manuscript, including in the title, to reflect the descriptive and correlational nature of the study.
Response 2: To better reflect the title and purpose of the study, it has been revised as "Association Between Work Stress and Eating Behavior: A Study among Academicians in Türkiye"
Materials and Methods:
Comments 3: Study Design: The section should begin by explicitly stating the type of study conducted (e.g., observational, descriptive, cross-sectional), which is standard in this section.
Response 3: The first sentence of the Study Design section is written to express the type of study being conducted.
Comments 4: Participants: A specific subsection describing the participants is still missing. This section should include detailed information on sample characteristics and explain how anonymity and data confidentiality were ensured.
Response 4: Information about the participants has been added.
Comments 5: Assessment Tools: This section requires restructuring and clearer writing. I recommend dividing it into three subsections:
- Ad hoc socio-demographic questionnaire
- German Workplace Stress Scale (GWSS)
- Adult Eating Behaviour Questionnaire (AEBQ). The manuscript still does not clearly define what the four "food approach" and the three "food avoidance" subscales are, nor what they imply behaviourally. This clarification is essential.
An appendix should be included with sample items or a full list of questions, especially for the ad hoc questionnaire.
Within the descriptions of the standardised instruments, the authors should provide brief information on their psychometric properties (e.g., validity and reliability). This content should be incorporated into the subsections corresponding to each instrument.
Response 5: As per the reviewer’s suggestion, the 'Assessment Tools' section has been presented in three parts. Under the heading 'Ad hoc Socio-demographic Questionnaire', details regarding the questions included in the survey have been added. Information about the subscales has been included under the 'Adult Eating Behaviour Questionnaire (AEBQ)' section. Validity and reliability information is provided solely within the context of the scale descriptions.
Comments 6: Reliability of Questionnaires: This section is unnecessary in its current form. It is not required to explain the general meaning of Cronbach’s alpha; instead, the authors should report the specific alpha values obtained for each instrument in the current study, and this information should be included in the relevant instrument subsections.
Response 6: The section 'Reliability of Questionnaires' has been removed from the text.
Results:
Comments 7: There remains a lack of explanation regarding what higher or lower scores in certain subscales—such as "Food Fussiness" or "Food Responsiveness"—indicate in behavioural terms. Although the authors claim to have added an interpretation to Table 3 and the scale description section, what has actually been included is a speculative interpretation of the results, which belongs in the conclusion, not in the results section. Without a proper explanation of the scale constructs and score meaning, the results are difficult to interpret.
Response 7: AEBQ score evaluation information is explained under the heading "2.2.3. The Adult Eating Behavior Questionnaire (AEBQ-TR)".
Comments 8: The sentence “the average eating behaviour scores of male are lower than those of female” remains ambiguous. As noted previously, it is unclear whether this means women eat more, eat differently, or exhibit different behavioural patterns. Without a clear explanation of what high or low AEBQ scores indicate, the significance of this result remains unclear.
Response 8: AEBQ score evaluation information is explained under the heading "2.2.3. The Adult Eating Behavior Questionnaire (AEBQ-TR)".
Comments 9: Moreover, the sentence “as the general work stress score increases, the AEBQ score increases” is not meaningful without specifying which dimensions of each scale are involved.
Response 9: The sentence "In other words, statistically weak but significant relationships were identified between overall work stress scores and the total score of the AEBQ-TR, as well as the subscales of Emotional Over-eating and Hunger + Food Responsiveness." has been changed to.
Comments 10: Regarding statistical analysis, the previous review noted: “The chi-square test is used to analyse associations between categorical variables. However, the GWSS and AEBQ are not inherently categorical. The manuscript does not specify how the continuous total scores from these instruments were categorised (e.g., low/medium/high) to allow for chi-square analysis.” The authors now claim that the chi-square test was only used in the context of the post-hoc power analysis. However, the information presented in Table 4 clearly includes chi-square comparisons across academic status groups. Thus, if categorical variables derived from continuous scores were used, the method for this categorisation must be explained. Otherwise, the table and corresponding text (lines 249–275) should be removed.
Response 10: The statistical analysis reported in Table 4 is the Kruskal-Wallis H test. It is shown as X2 because it is notated in some literature. The notation of the Kruskal-Wallis’s test has been corrected in the text.
Comments 11: The authors state that “the study focused on descriptive and correlational findings in line with its initially determined purpose”. However, this is not reflected consistently across the manuscript, starting with the title and continuing through the aims, analyses, and conclusions.
Response 11: Edits have been made in the text to ensure consistency between the purpose, analysis and results of the study.
Comments 12: Subsections were suggested for the Results section, but only two have been included, and neither aligns well with the main aim of the study (i.e., the relationship between occupational stress and eating behaviours).
Response 12: As indicated in the title, the main aim of our study is to examine the relationship between work stress and eating behaviors among academicians in Türkiye. In this context, the results are presented under subheadings that consider potential sources of stress such as gender, academic title, and promotion-related challenges. These subheadings were structured to reflect the study’s aim and the relationships between the applied scales."
Comments 13: There is still redundancy between the narrative and the tables. For example, "reasons to skip meals" appear both in Table 2 and in the main text. Similarly, Figure 1 and Table 3 seem to convey overlapping information.
Response 13: Editing has been done to minimize repetitions.
Comments 14: Typographic conventions should be observed—for instance, the letter n should appear in italics
Response 14: The arrangement has been made according to typographical rules.
Comments 15: In Table 4, the title is misleading. What is presented are differences in certain variables according to academic status, not associations between variables.
Response 15: The name of Table 4 has been changed upon your suggestion.
Comments 16: In Table 1, the inclusion of age and BMI in columns labelled "n" and "%" is not appropriate. Besides, these values are already reported in the text, and BMI is not even used in the analyses, so these rows could be removed.
Response 16: Age and BMI information has been removed from Table 1 upon your suggestion.
Discussion:
Comments 17: In this section, the authors should support their actual findings with studies specifically addressing the relationship between occupational stress and eating behaviour within academic settings, which is the stated focus of the study—or, at the very least, in other workplace environments. Studies involving adolescents or university students do not appear particularly relevant in this context. The study involving nurses might be more applicable, although stress in healthcare professions (e.g., among doctors and nurses) is likely not comparable to that experienced in academic settings.
In any case, the purpose of the discussion is not to present a list of unrelated studies—some more relevant than others—but rather to interpret and contextualise the results obtained, using the literature to shed light on the findings. The current discussion reads as the previous version with additional studies inserted, yet with no clear connection to the content that follows. As such, it lacks coherence and does not adequately engage with the study’s own results.
Response 17: We truly appreciate your valuable feedback. Our study was not designed to address occupational stress in a broad or generalized context, but rather to explore the relationship between work-related stress experienced by academicians and their eating behaviors. During our literature review, we did not come across any studies directly examining this relationship in the context of academic rank and gender. Based on your suggestions, we carefully revised the discussion section to better reflect the impact of work stress on eating behaviors among academicians and the potential influence of gender. Thank you once again for your thoughtful comments, which have played an important role in improving the overall quality of our work.
Conclusions:
Comments 18: The conclusions drawn are not supported by the results. The findings indicate: A low positive correlation between work stress and emotional overeating; A moderate correlation between work stress and hunger/food responsiveness; and no significant correlation between work stress and other eating behaviours (enjoyment of eating, emotional undereating, picky eating, or satiety sensitivity). Yet, the conclusion states: “This study reveals the effect of stress on eating habits and how this effect differs according to gender; it also shows that as academic title increases, significant changes are observed in stress levels and related eating habits scores.” This interpretation is problematic for several reasons:
The word "effect" suggests causality, which the study design does not support.
There does not even appear to be any statistically significant or substantively meaningful correlation between the variables."
The moderating role of gender or academic status has not been tested — the study only examined whether there are group differences in stress or eating behaviour scores separately.
The conclusion extrapolates beyond what the data support.
Response 18: The conclusion section has been revised in line with your suggestions.